# Interrelation Between Pathoadaptability Factors and Crispr-Element Patterns in the Genomes of *Escherichia coli* Isolates Collected from Healthy Puerperant Women in Ural Region, Russia

**DOI:** 10.3390/pathogens13110997

**Published:** 2024-11-14

**Authors:** Yulia Mikhaylova, Marina Tyumentseva, Konstantin Karbyshev, Aleksandr Tyumentsev, Anna Slavokhotova, Svetlana Smirnova, Andrey Akinin, Andrey Shelenkov, Vasiliy Akimkin

**Affiliations:** 1Central Research Institute of Epidemiology, Rospotrebnadzor, Novogireevskaya Str., 3a, 111123 Moscow, Russia; mihailova@cmd.su (Y.M.);; 2Federal Scientific Research Institute of Viral Infections «Virome», Letnyaya Str., 23, 620030 Ekaterinburg, Russia; smirnova_ss@niivirom.ru

**Keywords:** *Escherichia coli*, WGS, pathogenic potential, antibiotic resistance genes, virulence factors, CRISPR/Cas system, pathoadaptability

## Abstract

*Escherichia coli* is a commensal and opportunistic bacterium widely distributed around the world in different niches including intestinal of humans and animals, and its extraordinary genome plasticity led to the emergence of pathogenic strains causing a wide range of diseases. *E. coli* is one of the monitored species in maternity hospitals, being the main etiological agent of urogenital infections, endometriosis, puerperal sepsis, and neonatal diseases. This study presents a comprehensive analysis of *E. coli* isolates obtained from the maternal birth canal of healthy puerperant women 3–4 days after labor. According to whole genome sequencing data, 31 sequence types and six phylogenetic groups characterized the collection containing 53 isolates. The majority of the isolates belonged to the B2 phylogroup. The data also includes phenotypic and genotypic antibiotic resistance profiles, virulence factors, and plasmid replicons. Phenotypic and genotypic antibiotic resistance testing did not demonstrate extensive drug resistance traits except for two multidrug-resistant *E. coli* isolates. The pathogenic factors revealed in silico were assessed with respect to CRISPR-element patterns. Multiparametric and correlation analyses were conducted to study the interrelation of different pathoadaptability factors, including antimicrobial resistance and virulence genomic determinants carried by the isolates under investigation. The data presented will serve as a valuable addition to further scientific investigations in the field of bacterial pathoadaptability, especially in studying the role of CRISPR/Cas systems in the *E. coli* genome plasticity and evolution.

## 1. Introduction

*Escherichia coli* is one of the most divergent and widespread species that can behave as a commensal in the human gastrointestinal tract and persists in water and soil regardless of the host. At the same time, it can behave as a pathogen causing different types of diseases both in and outside of a human or animal host’s digestive system [1]. The remarkable and complex plasticity of the *E. coli* genome contributes to the formation of the pathogenic potential. Thus, as a result of taking up and accumulating pathogenic factors, multidrug-resistant high-risk clones capable of causing a wide range of diseases in humans in a certain biotope/system can emerge [2,3].

According to the combination of special genetic markers, *E. coli* can be subdivided into eight phylogenetic groups: A, B1, B2, C, D, E, F, and G. Such phylogroup classification of *E. coli* is useful for comparative analysis of serogroup, virulence, and resistance traits, as well as diversity assessment of *E. coli* populations within various hosts and environments [4,5].

The pathogenic potential of *E. coli* is implemented by the presence of various virulence factors. There are different *E. coli* pathotypes represented by a group of clones that share a certain set of specific virulence determinants [2]. The most common *E. coli* ST131 clone from phylogenetic group B2 is the predominant clone of high-risk worldwide, and recently another ST1193 clone was recognized as an emerging high-risk clone belonging to the same group. High-risk *E. coli* clones are spreading very quickly, which leads to their existence in different niches, in human and animal intestinal tracts, and persistence in the environment [3]. Moreover, phylogenomic approaches have shown that only four sequence types (STs) are responsible for extraintestinal infections (ST131, ST73, and ST95 of a phylogroup B2, and ST69 of a phylogroup D), which were always studied in epidemiological surveillance investigations and were therefore named “the big four ExPEC (extraintestinal pathogenic *E. coli*) clones” [6].

Most *E. coli* strains are characterized by possessing either type I-E or I-F CRISPR/Cas systems. These findings can explain an evolutionary interrelation between CRISPR and pathogenicity in *E. coli* [7]. Taking into account the continuously accumulating data, it becomes apparent that adaptive immunity is not the only role of CRISPR/Cas systems. It was shown that the expression of many bacterial genes affecting the virulence and group behavior of pathogenic bacteria is regulated by these systems. Additionally, CRISPR/Cas participates in DNA repair and accelerates the evolution of the genomes [8]. Other studies indicated an association between the presence of the CRISPR sequence and decreased antibiotic resistance, thus suggesting that the presence of CRISPR limits the adaptability of the microorganism [7,9].

It is noteworthy that most genome sequencing projects emphasize the analysis of multidrug-resistant clones, clinically relevant pathogens, or epidemiological outbreaks. However, there is a lack of genomic data on opportunistic *E. coli* populations obtained from healthy patients, namely, puerperant women, who take a specific place in maternity care facilities. They are not patients with some kind of pathology, but, at the same time, they are “healthy patients” with an open system (maternal birth canal) or surgical suture (after caesarian section), which are susceptible to infections in hospital conditions. Women who give birth by cesarean section labor are estimated to have a 5-fold to 20-fold risk of bacterial infection in comparison to women who give birth vaginally [10]. Regardless of the labor type, women in the early postpartum period face transient immunodeficiency, and decreased activity of local tissue immunity, which increases their sensitivity to bacterial infection [11]. At the same time, *E. coli* bacteria play a major role in the etiology of puerperal sepsis [12].

As a part of our ongoing molecular surveillance program, here we provide a detailed investigation of *E. coli* isolates collected from maternal birth canal discharge of puerperant women in two perinatal centers of Ekaterinburg, Russia (referenced as ‘Crie-Pu’ isolates below). We used whole genome sequencing (WGS) to characterize *E. coli* isolates as a dominating bacteria collected. The entire collection of the strains was examined in terms of population structure, phenotypic and genotypic profiles of antimicrobial resistance, virulence factors, plasmid replicons, and analysis of CRISPR-elements patterns. Information on resistance and virulence genes and correlation analysis is an important tool for epidemiological studies in assessing the pathogenic potential and total pool of important determinants in a population of opportunistic *E. coli* and monitoring the emergence of new clinically and epidemically significant resistance and virulence phenotypes.

## 2. Materials and Methods

### 2.1. Isolate Collection

The study was conducted by continuous sampling method in the postpartum department of two Perinatal Centers in the Ural region. It included 100 and 130 clinically healthy puerperant women on the 3rd–4th day after labor immediately before discharge from the department of municipal and regional perinatal centers, correspondingly. Nurses of the departments collected biological material from the maternal birth canal with the informed consent of the women. The samples were collected using sterile disposable probes followed by preliminary seeding on a transport medium. Isolation of pure bacterial cultures was carried out by seeding on solid nutrient media (Endo Agar) with subsequent species identification based on cultural-morphological, biochemical, and antigenic properties. The species for all isolates under study were identified by time-of-flight mass spectrometry (MALDI-TOF MS) using the VITEK MS system (bioMerieux, Marcy-l’Étoile, France).

Sixty-two bacterial cultures of opportunistic pathogens were obtained from the municipal perinatal center, including 10 *E. coli* isolates. One hundred opportunistic bacterial pathogens including 43 *E. coli* isolates were collected from the patients of the regional perinatal center. The age of patients from whom *E. coli* samples were isolated (*n* = 53) ranged from 19 to 42 years with a median equal to 29 (Appendix A).

### 2.2. Determination of Antibiotic Susceptibility

The antibiotic susceptibility was evaluated by the disc diffusion method using the Mueller–Hinton medium (bioMerieux, Marcy-l’Étoile, France) and disks with antibiotics (BioRad, Marnes-la-Coquette, France), and by the boundary concentration method on VITEK2 Compact 30 analyzer (bioMerieux, Marcy-l’Étoile, France). The isolates were tested for their susceptibility/resistance to the following antimicrobial drugs: imipenem, amoxicillin, amoxicillin/clavulanic acid, amikacin, gentamicin, levofloxacin, ciprofloxacin, cefotaxime, cefixime, ceftriaxone, and cefuroxime. The EUCAST clinical breakpoints version 12 was used to interpret the results obtained (https://www.eucast.org/clinical_breakpoints/, accessed on 20 December 2022).

### 2.3. Whole Genome Sequencing

Genomic DNA was isolated using a DNeasy Blood and Tissue kit (Qiagen, Hilden, Germany) and passed to the paired-end library preparation with Nextera™ DNASamplePrepKit (Illumina^®^, SanDiego, CA, USA). Whole genome sequencing (WGS) of 53 isolates was conducted on the Illumina^®^ NextSeq2000 platform (Illumina^®^, San Diego, CA, USA). Assemblies were obtained using SPAdes versions 3.15.2 and 3.15.4 and were uploaded to NCBI Genbank under the project number PRJNA1151703.

### 2.4. Data Processing

The genomes assembled were processed using the custom pipeline described earlier by us [13,14]. In brief, for all isolates, we performed MLST typing using the Achtman 7 Gene MLST scheme with the profiles presented in Enterobase (https://enterobase.warwick.ac.uk/species/index/ecoli, accessed on 21 May 2024), and serotyping was made with SerotypeFinder 2.0 (https://cge.food.dtu.dk/services/SerotypeFinder/, accessed on 21 May 2024). Phylogroups were assigned based on the data from Clermont et al. [15]. We used the Resfinder 4.0 database with default parameters for antimicrobial gene identification (http://genepi.food.dtu.dk/resfinder, accessed on 22 May 2024). Plasmid sequences were revealed and typed using PlasmidFinder 2.1 with default parameters (https://cge.food.dtu.dk/services/PlasmidFinder/, accessed on 26 May 2024). Virulence factors were revealed by searching in VFDB (http://www.mgc.ac.cn/VFs/main.htm, accessed on 25 May 2024).

CRISPRCasFinder version 4.2.20 with default parameters [16] was used to identify the presence of CRISPR/Cas systems and spacers in the genomes analyzed. The data on CRISPR-elements in 3210 reference isolates were obtained from the CRISPRCas database (https://crisprcas.i2bc.paris-saclay.fr/MainDb/StrainList, accessed on 3 July 2024).

The analysis of spacers in CRISPR arrays of Crie-Pu *E. coli* isolates with putative CRISPR/Cas systems was performed by Web BLAST^®^. The spacers were identified and downloaded from the CRISPRDetect web tool [17] (http://crispr.otago.ac.nz/CRISPRDetect/predict_crispr_array.html, accessed on 10 July 2024). CRISPRDetect FASTA sequences of Crie-Pu *E. coli* spacers were uploaded to Web BLAST^®^ blastn suite (https://blast.ncbi.nlm.nih.gov, accessed on 10 July 2024) and analyzed using default parameters of the MegaBLAST algorithm. In addition, CRISPRDetect FASTA sequences of the spacers were uploaded to the CRISPRTarget web tool [18,19] (http://crispr.otago.ac.nz/CRISPRTarget/crispr_analysis.html, accessed on 10 July 2024).

Phylogenetic analyses were conducted using the Maximum Likelihood (ML) method in MEGA X [20] as described earlier [21].

To assess the differences between the distribution of antimicrobial resistance (AMR) genes, virulence factors, and plasmid replicons in the studied *E. coli* population, Fisher’s Exact Test was used (https://www.langsrud.com/fisher.htm, accessed on 16 July 2024). Corresponding data analysis and graphing were performed using Prism version 9 (GraphPad Software, San Diego, CA, USA).

## 3. Results

### 3.1. Isolate Typing

The 53 isolates of *E. coli* belonged to 31 different STs without a significant prevalence of any individual one. Most numerous were ST69 (n = 8), ST73 (n = 5), and ST131 (n = 6), while 29 different STs were represented only by less than four isolates or even by a single one. Phylogroup classification demonstrated that *E. coli* isolates were distributed in six diverse phylogroups—A, B1, B2, C, D, and F, with the prevalence of B2 represented by 32 samples belonging to 18 STs (Appendix A). The phylogroup D combined nine isolates of ST69 and ST349 (single sample). A and B1 phylogroups consisted of four and five isolates, respectively, while C and F were represented by a single sample each (Appendix A). The exact phylogenetic group was not identified for three isolates (Crie-Pu1335, 1340, and 1370), the last of which carried boundary traits of B2 and F phylogroups. Additionally, twenty different O serogroups were distinguished, O50 (15%), O15 (13%), O16 (7.5%), and O75 (9%) were the most frequent among the isolates under investigation. O-serogroup diversity is described in more detail in the section “Virulence genes” as long as both O-serogroup antigen and virulence factors are essential for the pathogenicity assessment of *E. coli*.

### 3.2. CRISPR Element Distribution of the Crie-Pu E. coli Isolates

Approximately 68% of the *E. coli* genomes available in the CRISPRCasdb are predicted to harbor active CRISPR/Cas systems (https://crisprcas.i2bc.paris-saclay.fr/MainDb/StrainList, accessed on 3 July 2024). High variability of CRISPR/Cas systems was observed along with previously known diversity and plasticity of *E. coli* genomes. This is the first work that shows a thorough description of CRISPR/Cas elements in Russian *E. coli* isolates of clinical origin (listed in Appendix A) collected from healthy patients, in comparison to the strains of the same species that were available from public databases (https://crisprcas.i2bc.paris-saclay.fr/MainDb/StrainList, accessed on 3 July 2024).

A total of 26.4% (14 out of 53) of the studied *E. coli* isolates harbored neither CRISPR array nor cas cassette. Six isolates (11.3%) from the set possessed confirmed CRISPR arrays but lacked cas cassettes, and thirty-three samples (62.3%) carried different putative CRISPR/Cas systems containing cas cassettes (Appendix A). It should be noted that a similar distribution of CRISPR arrays and CRISPR/Cas systems containing cas cassettes was observed for our *E. coli* isolates and the 3210 reference *E. coli* isolates available in the CRISPRCas database (https://crisprcas.i2bc.paris-saclay.fr/MainDb/StrainList, accessed on 3 July 2024). Namely, 2470 and 2193 isolates were characterized as carrying CRISPR arrays and cas cassettes, respectively, while 740 strains did not have any CRISPR/Cas elements.

Among 33 Crie-Pu *E. coli* isolates carrying putative CRISPR/Cas systems containing cas cassettes, 19 carried the Type I-E CRISPR/Cas system (57.5%) and 14 carried the Type I-F CRISPR/Cas system (42.4%). It should be noted that Type I-E CRISPR/Cas systems are found more frequently (*p* ˂ 0.05) in the group of reference *E. coli* isolates available in the CRISPRCas database (https://crisprcas.i2bc.paris-saclay.fr/MainDb/StrainList, accessed on 3 July 2024), while Type I-F CRISPR/Cas systems occurred much more frequently (*p* ˂ 0.00001) in the experimental group of Crie-Pu isolates.

Crie-Pu isolates bearing neither a CRISPR array nor a cas cassette belonged to eight different sequence types, the predominant being ST131 and ST73—five and three Crie-Pu isolates without cas cassettes belonged to these STs, respectively (Figure 1). Crie-Pu isolates with confirmed CRISPR arrays, but without cas cassettes, belonged to 5 different sequence types, the predominant being ST73 possessed by two Crie-Pu isolates (Figure 1).

Crie-Pu isolates with Type I-E CRISPR/Cas systems belonged to 12 different sequence types with the dominance of ST69 (eight out of 19 Crie-Pu isolates) (Figure 1). The isolates with Type I-F CRISPR/Cas systems belonged to nine different sequence types, the predominant being ST141 (four isolates), ST1993, and ST80 (two isolates each) (Figure 1).

All Crie-Pu isolates bearing neither a CRISPR array nor cas cassette belonged to the phylogenetic group B2 (Figure 2), while the isolates with confirmed CRISPR arrays, but without cas cassettes, belonged to three different phylogenetic groups—B2 (four isolates), A, and D (one isolate each) (Figure 2).

Crie-Pu isolates with Type I-E CRISPR/Cas systems belonged to six different phylogenetic groups—D (eight isolates), B1 (4), A (3), F, C, and B2/F, while the isolate Crie-Pu1335 was not assigned to any known phylogroup (Figure 2). Almost all isolates with Type I-F CRISPR/Cas systems belonged to the B2 phylogenetic group, and only Crie-Pu1340 was not assigned to any group (Figure 2).

At the same time, almost all of the Type I-E CRISPR/Cas systems (18 out of 19) consisted of eight genes encoding Cas1, Cas2, Cas3, Cas5, Cas6, Cas7, Cse1, and Cse2. Crie-Pu1332 Type I-E isolate carried four *cas* genes—*cas3*, *cse1*, *cas1*, and *cas2*, and Crie-Pu1252 Type I-E isolate carried an additional *cas3* gene.

According to the phylogenetic analysis of the full-length Type I-E *cas* gene sequences, ST69 isolates formed a separate clade on phylogenetic trees, while the topology of the Crie-Pu isolates belonging to the other STs differed slightly between the phylogenetic trees (Figure 3).

The CRISPR/Cas loci of most Type I-F Crie-Pu isolates (13 out of 14) consisted of six genes encoding Cas6/Csy4 endoribonuclease, three Csy proteins (Csy3, Csy2, and Csy1), Cas3f helicase/RNase, and Cas1f endonuclease located in the vicinity of CRISPR arrays, except for the Crie-Pu1367 isolate, which carried only *cas6*, *csy1*, *csy2*, and *csy3* genes.

Phylogenetic analysis of the Type I-F *cas* gene sequences showed that separate clades on phylogenetic trees were formed by (i) *E. coli* Crie-Pu ST141 and ST1993 isolates and (ii) *E. coli* Crie-Pu ST80 isolates. It is worth noting that the isolates belonging to other genetic lines (STs) did not form separate clades and were dispersed throughout the phylogenetic trees (Figure 4).

### 3.3. Susceptibility to Antibiotics

Phenotypic antimicrobial susceptibility testing showed that 28 *E. coli* isolates (53%) were susceptible to all antibiotics in the panel used (Appendix A). Phenotypic resistance to three antimicrobial compounds of different groups (aminoglycosides/penicillins, fluoroquinolones, and cephalosporins) was detected in two isolates (Crie-Pu 1299 and 829, correspondingly). Resistance to two antibiotics of different groups simultaneously (cephalosporins in combination with aminoglycosides, penicillins, or fluoroquinolones, as well as aminoglycosides and fluoroquinolones) was detected in 11 isolates. The remaining 12 isolates were resistant to only one of the antibiotics used in the panel. The identified phenotypes were not associated with the type of labor (cesarean section or vaginal birth), as well as with the prescribed antibiotic therapy. However, it should be noted that the multidrug-resistant isolates (Crie-Pu1299 and 829) were identified in two puerperant women with a cesarean section and one with a complicated vaginal birth. In these two cases, antibiotic therapy was prescribed (Appendix A).

### 3.4. Antimicrobial Resistance Genetic Determinants

In silico searching for AMR determinants revealed the genes and gene clusters conferring resistance to aminoglycosides, beta-lactams, chloramphenicol, fluoroquinolones, trimethoprim, fosfomycin, macrolides, sulfonamides, and tetracycline.

The genomes of seven isolates of 28 susceptible to all antimicrobial compounds were characterized by the presence of 1 to 7 AMR genes including carbapenemase-encoding *bla_DHA-1_* and extended-spectrum β-lactamase (ESBL)-encoding genes (*bla_CTX-M-15_* and *bla_TEM-1C_*) (Appendix A).

The genomes of all remaining antibiotic-resistant isolates, except for four isolates belonging to phylogroup B2, contained ESBL- and other β-lactamase encoding genes: *bla_TEM_* (n = 19), *bla_CTX-M_* (n = 15), *bla_DHA-1_* (n = 2), and *bla_OXA-1_* (n = 1). Two isolates of the B2 phylogroup (Crie-Pu1235 and 1290) combined two β-lactamase genes of different types in their genomes (*bla_CTX-M-27_* + *bla_TEM-1B_* and *bla_CTX-M-15_* + *bla_OXA-1_*, respectively). It is noteworthy that both strains belonged to ST131, but had different serotypes O16:H5 and O25:H4 (Appendix A). The diversity of identified alleles of the epidemically significant ESBL gene *bla_CTX-M-3_*, *_-15_*, and *_-27_* draws attention since these genes were revealed in almost a third of the studied strains from a short period of research within the same perinatal center and in a small group of puerperant women.

Most of the isolates of phylogroup B2 (14 out of 22) lacked genes responsible for resistance to aminoglycosides, chloramphenicol, macrolides, and sulfonamides. Four isolates of that group carried only a single gene determining resistance to chloramphenicol (two isolates), tetracycline, or trimethoprim (Appendix A).

All isolates of the phylogroup D were characterized by the presence of 1–4 genes that determined resistance to aminoglycosides and one or two genes encoding resistance to trimethoprim. We also marked a combination of *aadA5* (aminoglycoside) and *dfrA17* (trimethoprim) resistance genes in *E. coli* genomes simultaneously. The only isolate (Crie-Pu1256) from this phylogroup and the entire studied set was characterized by the presence of the *fosA* providing resistance to fosfomycin (Appendix A).

Antimicrobial resistance genes were observed more frequently (*p* = 0.0058) in the group “Type I-E” *E. coli* isolates when compared to the “Type I-F” isolates (Figure 5).

### 3.5. Virulence Genes

*E. coli* isolates under investigation included a diverse repertoire of virulence-associated genes with 24 genes detected in each of the 53 isolates, namely, the multidrug efflux pump subunit gene *acrB4*; the allantoinase gene *allB*; the intimin-like adhesin gene *fdeC*; the outer membrane protein gene *ompA*; the enterobactin iron acquisition system genes *entABCES*; the ferric uptake genes *fes* and *fur*; the phosphogluconate dehydrogenase gene *gndA*; the invasion genes *ibeB*, *C*; the transcriptional regulatory protein genes (*cgs* operon, *phoP*, *pmrA*, *rcsB*, *rpoS*); and the fimbrial chaperone genes (*yagV-Z/ecpA-E*).

Additionally, more than 200 virulence genes were present variably, according to phylogroup and serotype of the isolates under investigation (Appendix A). Genes involved in adhesion, iron acquisition, immune evasion, and toxins were widespread. Since the virulence profiles of the isolates were very heterogenic, we will describe them by focusing on the main functional groups mentioned above.

Adhesins represent the molecules involved in signaling pathways between bacteria and host cells thereby stimulating tissue colonization and invasion [22]. This is the most important pathogenicity-related factor in *E. coli*. The most common adhesion determinants in our sample set were *fim* genes encoding type 1 fimbriae. All isolates except one (Crie-Pu1338) harbored this gene cluster. The main part of the isolates of B2 and D phylogroups carried genes of P-fimbriae (*pap*). The isolates of serogroups O25 and O50 had the most complete gene set composing this cluster. The genes associated with mannosoresistant pili (*sfa* and *foc*) were observed only in B2 isolates mainly belonging to serogroups O6, O18, and O50 (Appendix A).

Iron acquisition is a significant property facilitating bacterial survival during the infection process. It is also necessary for general growth, fitness, and electron transfer during cellular respiration [23]. The main players of iron uptake in bacterial cells are different types of siderophores, namely, aero, entero-, yersiniabactins, and salmochelin. As to the *E. coli* strains of our research, the full aerobactin operons *iucABCD* and *iutA* were present in 43.4% (n = 23) of the isolates. In the dominant phylogroup B2, these genes were associated with O16, O18, and O25 serotypes. A ferric enterobactin cluster (*fepA-G*) was observed in all isolates except Crie-Pu1367 (ST141-B2-O50:H6). The ferric yersiniabactin uptake receptor *fyuA* was not found in only six isolates. Four of them belonged to A (n = 2) and B1 (n = 2) phylogroups, and the rest—to C and D phylogroups. The iron-regulatory proteins *irp1* (n = 45, 85.0%) and *irp2* (n = 44, 83.7%) were present in the same fraction, and the complete yersiniabactin siderophore operon *ybtAEPQSTYX* was additionally presented in these isolates. The salmochelin siderophore system encoded by *iroBCDEN* was revealed in 90.0% (n = 48) of the isolates belonging to all phylogenetic groups except for Crie-Pu1239 (St59-O1:H7) of the F phylogroup. B2-ST131-O16:H5 isolates (n = 4) were characterized by the absence of salmochelin genes, and the same iron uptake genes profile was exhibited by ST131-O25:H4 isolates (n = 2), and one of each isolate from ST404 and ST1193 belonging to the same serotype (O75:H5). Most of the B2 isolates (n = 16) were characterized by the absence of aerobactin operone, predominantly, and these samples mostly referred to O6:H1, O50:H6, and O75:H7 serotypes (Appendix A).

Toxins play an important role in infections of different localizations since they contribute to the spread of bacteria in tissues, increasing cytotoxicity, resistance to neutrophils, as well as damage and disruption of host cell metabolism, leading to a biotope environment more favorable for *E. coli* [24]. Toxin genes were not revealed in two isolates with unidentified phylogroups (Crie-Pu1335 and 1340); in two samples of A and all isolates of B1 phylogroups (n = 4); in a single strain of C1 (Crie-Pu1330); and in separate isolates of B2 and D phylogroups (n = 5 and n = 2, correspondingly). The most frequently identified toxin genes were *cnf1* (cytotoxic necrotizing factor) and *hlyA-D* (hemolysin). The combination of these genes was more common for B2 isolates, especially, for serogroups O50 and O6 (Appendix A). Several isolates belonging to A, B2, D, and F phylogroups harbored serine protease autotransporters genes (*pic* and *sat*) and the enterotoxin determinant *senB*. The combination with maximal numbers of toxin determinants was identified in three isolates of the B2 phylogroup, with two of them (Crie-Pu1235 and 1290, both ST131) belonging to O16:H5 and O25:H4 serotypes, correspondingly, and the isolate Crie-Pu1304-ST73-O18ac:H1 (Appendix A).

Protectins defend bacteria from the host immune system and from various unfavorable conditions. In particular, protectins include bacterial capsules, outer membrane proteins, and lipopolysaccharide components. Most of the *E. coli* isolates under investigation (n = 38, mainly the isolates from the B2 and D phylogroups) carried the *kps* cluster, determining the synthesis of a polysialic acid capsule. Some of these isolates additionally harbored the *neu* cluster responsible for polysialic acid production. Only ten isolates of the B2 phylogroup (mainly belonging to O50 and O6 serotypes) carried the *tcpC* gene involved in the suppression of innate immunity (Appendix A).

We observed insignificant differences in virulence gene numbers for the *E. coli* isolates vs. CRISPR-element patterns. Meanwhile, virulence factor genes were found more frequently (*p* = 0.0412) in the group “Type I-F” *E. coli* isolates when compared to the “Type I-E” isolates (Figure 6).

### 3.6. Plasmid Replicons of Crie-Pu E. coli Isolates

Since plasmids are very important vehicles of antibiotic resistance genes [25], we performed a search of plasmid sequences in the WGS data obtained. The list of plasmid replicons identified in the isolates is shown in Appendix A. Plasmid sequences were revealed in 50/53 (94%) isolates analyzed. In total, 24 different types of plasmid replicons belonging to 15 varieties of Inc-type plasmids and nine varieties of Col-type plasmids were found (Appendix A). IncFIB and Col156 plasmids were the most frequently carried replicons in our collection, revealed in 27 and 18 Crie-Pu isolates, respectively. Col- or Inc-type plasmid replicons were separately carried by nine and seven *E. coli* isolates, respectively, while thirty-seven isolates under investigation included the replicons of both types. The total replicon number usually was between 1 and 4, while the maximum number of plasmid replicons belonging to different types in one *E. coli* genome reached eight (Crie-Pu1331) (Appendix A).

However, in order to obtain reliable plasmid sequences, long-read sequencing is usually required, which we plan to perform in the future.

According to CRISPR-element sets of the studied isolates, plasmid replicons were observed more frequently (*p* = 0.0333) in the group “Type I-E” *E. coli* isolates when compared to the “Type I-F” isolates (Figure 7).

### 3.7. CRISPR Arrays of the E. coli Isolates

The CRISPR array spacers number did not differ significantly between “Type I-E” and “Type I-F” *E. coli* isolates, but the number of spacers in both groups was significantly higher than in “CRISPR/No Cas” (Figure 8).

CRISPR arrays of Type I-E Crie-Pu *E. coli* isolates consisted of 450 spacers (189 unique spacers and 261 repeating spacers with 48 unique spacers among them). 71.1% of the analyzed spacers were identified as *E. coli* CRISPR spacers (e.g., ‘*Escherichia coli* strain C2-7 CRISPR repeat region’) using the MegaBLAST algorithm of Basic Local Alignment Search Tool (BLAST^®^, National Library of Medicine). Only 4.2% of spacers were identified as phage sequences (e.g., ‘Bacteriophage sp. isolate 4198_46168, partial genome’), and 49.3% of spacers targeted plasmids (e.g., ‘*Escherichia coli* strain KE47 plasmid unnamed1, complete sequence’) (Appendix A). The rest (around 14.2%) were self-targeting spacers (Appendix A).

In total, 224 spacers were found in Type I-F Crie-Pu *E. coli* CRISPR arrays, with 75 of them being unique and 149 being repeating (with 39 unique spacers among the repeating ones). In addition, 61.6% (138 out of 224) were identified as spacers from known *E. coli* CRISPR arrays (e.g., ‘*Escherichia coli* strain 718 CRISPR1 repeat region’). Only five spacers (out of 224) targeted phage sequences (e.g., ‘*Salmonella* phage SW3, complete genome’), and ten spacers targeted plasmids (e.g., ‘*Escherichia coli* strain SCU-204 plasmid pSCU-204-5, complete sequence’) according to the analysis conducted using MegaBLAST algorithm of BLAST^®^ (National Library of Medicine) (Appendix A). The rest spacers (32.6%) targeted *E. coli* genomes (Appendix A).

Additionally, Type I-E and Type I-F CRISPR spacers were analyzed using the CRISPRTarget web service (http://crispr.otago.ac.nz/CRISPRDetect/predict_crispr_array.html accessed on 10 July 2024 [17]; http://crispr.otago.ac.nz/CRISPRTarget/crispr_analysis.html accessed on 10 July 2024 [18,19]). CRISPRTarget revealed specific targets (namely, ‘plasmids’, ‘phages’, ‘pathogenicity islands’ or combinations thereof) for 31 Type I-E and 26 Type I-F *E. coli* CRISPR spacers with 61.3% (19 out of 31) Type I-E spacers identified as targeting plasmids, while 50% (13 out of 26) Type I-F spacers were identified as targeting phages. It was shown that plasmid-targeting spacers were observed more frequently (*p* ≤ 0.0001) within Type I-E spacers, whereas phage-targeting spacers (*p* ≤ 0.001) were observed more frequently within Type I-F spacers (Figure 9).

### 3.8. Correlation Analysis

In order to find out the interrelation between pathoadaptability factors and CRISPR-element patterns, a correlation analysis was performed. First of all, the datasets “Antibiotic resistance genes count”, “Virulence factors genes count” “Plasmids count” and “Spacers count” for “No CRISPR/No Cas”, “CRISPR/No Cas”, “Type I-E”, and “Type I-F” groups of Crie-Pu *E. coli* isolates were normalized. Afterward, the nonparametric Spearman correlation was calculated, and correlation matrices were constructed. The resulting correlation matrices are presented in Appendix A.

In the “No CRISPR/No Cas” group of *E. coli* isolates, a moderate negative correlation (*p* = 0.025) was observed between the number of virulence factor genes and plasmids count (Table 1), while the number of AMR genes positively correlated with the number of plasmids in the “Type I-E” and “Type I-F” groups (Table 1).

## 4. Discussion

The work presented here focuses on whole genome analysis of *E. coli* strains isolated from the maternal birth canal discharge of puerperant women on the 3rd or 4th day after labor. We described our results through the prism of CRISPR-elements patterns to assess the pathogenic potential of the isolates under study. Our previous investigations showed specific features of multidrug-resistant bacteria in terms of their pathoadaptability, and genomic and phenotypic adaptations that promote bacterial survival under hospital conditions [21,26]. It is worth noting that there is a lack of data regarding the phenotypic and genotypic characterization of *E. coli* strains colonizing healthy women several days after labor.

The analysis of the population structure of the isolates showed a notable heterogeneity of the strains at genotype and phylogroup levels. The 53 studied samples belonged to 31 MLST-based sequence types and 20 O-serotypes. Phylogroup classification demonstrated that *E. coli* isolates under investigation were distributed in six phylogroups—A, B1, B2, C, D, and F, and more than half of the isolates (32, 60%) belonged to phylogroup B2. It was reported that most ExPEC strains derived from this group [27,28], and common ExPEC types were revealed in our sample set. These types included ST73 (three isolates), ST95 (one isolate), ST131 (six isolates), ST1193 (one isolate), and ST69 (phylogroup D, eight isolates). Interestingly, similar heterogeneity of *E. coli* isolates with the prevalence of the B2 phylogroup was described earlier for vaginal and endocervical strains collected from pregnant women [29,30] and cervix isolates of women in preterm labor [31]. It is also worth mentioning that four ST141 isolates were revealed in our study since their prevalence has recently increased in both extraintestinal and intestinal diseases [6]. In general, such a diversity of isolates collected from one department during a rather short period practically eliminates the possibility of nosocomial infection in this case.

In total, 62.3% of Crie-Pu *E. coli* isolates were characterized by the presence of the CRISPR/Cas systems. Commonly, such a high percentage of CRISPR/Cas systems is found in the microorganisms used in fermentation processes (e.g., starter cultures, probiotic cultures, and so on) [32,33]. CRISPR/Cas systems provide protection to these microorganisms from phage invasions that dramatically affect the quality of products [34] and represent one of the evolutionary mechanisms of adaptation and survival.

CRISPR/Cas systems Type I-E and I-F were found in 35.8% and 26.4% of Crie-Pu *E. coli* isolates, respectively. On the one hand, Type I-E CRISPR/Cas distribution is consistent with the one published earlier [35,36]. On the other hand, we found that Type I-E CRISPR/Cas systems are found more frequently (*p* ˂ 0.05) in the group of reference *E. coli* isolates available in the CRISPRCas database. Type I-F CRISPR/Cas systems were overrepresented (*p* ˂ 0.00001) in our collection of *E. coli* isolates compared to the data published earlier [35] and the fraction revealed in the reference group. This is due to the predominance of B2 isolates in our sample set since the Type I-F CRISPR/Cas system is a characteristic of B2 *E. coli* strains [37]. It is worth noting that the Type I-F CRISPR/Cas system is more active than Type I-E, so it possesses a higher potential association with pathogenicity due to its presumable affinity to genetic elements and low prevalence [7,38]. Interestingly, according to phylogenetic analysis of Type I-E and I-F *cas* genes, Crie-Pu *E. coli* isolates belonging to ST69, 141, and 80 formed separate clades on maximum-likelihood trees, thus indicating their greater divergence compared to reference isolates from the database.

Apparently, there are several factors contributing to bacterial pathoadaptability, such as the number of AMR genes, virulence factors, plasmids, and the presence of apparently functional CRISPR/Cas systems. In the present study, we demonstrated that the number of AMR genes and plasmid replicons was higher in *E. coli* isolates bearing Type I-E CRISPR/Cas system than in isolates with Type I-F systems, and this fact is consistent with the observation published earlier [39]. In 2019, Long et al. demonstrated the role of the Type I-F system in limiting the acquisition of AMR [37]. Interestingly, the majority of our *E. coli* isolates carrying the I-F system included fewer antimicrobial resistance genes compared to other isolates under investigation, but were characterized by a lower number of plasmid-targeting spacers than Type I-E Crie-Pu *E. coli* isolates. This fact is in contrast to the work of Aydin S. et al., who reported a much larger proportion of plasmid-specific spacers in *E. coli* isolates susceptible to antimicrobial drugs [39]. This observation suggests a more complex role of CRISPR/Cas systems in bacterial pathoadaptability and needs further investigation.

At the same time, in Type I-F *E. coli* isolates the number of genes encoding virulence factors was higher. Several virulence factors such as *sfa* (S fimbriae), *cnf1* (cytotoxic necrotizing factor 1), *focF* and *focH* (Fimbriae of serotype 1C), *iroB*, and *hlyABCD* (hemolysins) were seen more frequently in the group of Type I-F vs. Type I-E isolates. These virulence factors are frequently found in Uropathogenic *Escherichia coli* (UPEC), which represents the primary cause of UTIs globally [40] and is also associated with cystitis and pyelonephritis [41,42].

It is worth noting that the CRISPR/No Cas group of our Crie-Pu sample set was represented by six isolates only and they were characterized by an interrelation pattern of pathoadaptability factors similar to Type I-E *E. coli* isolates.

Furthermore, remarkable correlations were found between the pathoadaptability factors, such as the number of AMR genes, the number of virulence factors, and the number of plasmid replicons in the groups of Crie-Pu *E. coli* isolates with different types of CRISPR/Cas systems. Thus, for Crie-Pu *E. coli* isolates without CRISPR/Cas systems, we revealed the pattern of “more virulence genes-less plasmids” and vice versa. Another correlation pattern “more plasmids-more antibiotic resistance genes” and vice versa was observed in the “Type I-E” and “Type I-F” groups. This fact confers the potential association of Type I-E and Type I-F with *E. coli* pathogenicity [7] and may contribute to the pathoadaptability of *E. coli* with different CRISPR/Cas systems. Moreover, the presence of spacers with identities to phage and plasmid sequences in Crie-Pu *E.coli* strains indicates the defense role of CRISPR/Cas systems.

In our study, 14.2% of Type I-E spacers and 32.6% of Type I-F spacers were identified as self-targeting. It is known that *E. coli* CRISPR spacers have a statistically significant tendency to target hosts compared to phage genomes [43]. It is known that self-targeting spacers in CRISPR/Cas systems acquired from the host chromosome are involved in autoimmunity and cell death [44]. Moreover, such spacers play an important role in the mRNA degradation process making it possible to overcome host immune reactions [45]. In addition, self-targeting spacers may contribute to bacterial gene regulation and evolution [8,46]. It is assumed that CRISPR/Cas systems containing self-targeting spacers should be tightly regulated to maintain a balance between the risk of developing autoimmune reactions leading to cell death and the ability to resist phage invasions. These CRISPR/Cas systems still require further investigation [47].

## 5. Conclusions

Our whole genome-based analysis demonstrated that *E. coli* isolates from birth canal discharge of healthy puerperant women possessed a high potential for extraintestinal infections. This applies mainly to the isolates belonging to B2 and D phylogroups. We also observed colonization with *E. coli* strains from A, B1, C, and F phylogroups, which more commonly behave as commensals [48]. Additionally, the remarkable heterogeneity of the studied bacterial population was observed, which makes the nosocomial origin of *E. coli* infections unlikely. We believe that the data presented will contribute to further investigations in the field of bacterial pathoadaptability, and will serve as an important supplementary for the formation of new approaches for genomic epidemiology surveillance in clinical conditions, and in maternal care facilities, in particular. The assessment of antimicrobial resistance and virulence genetic determinants, plasmid replicons, etc. with respect to the variability of CRISPR elements for bacterial strains could be used as additional pathogenicity predictors and could facilitate the development of better prevention strategies against this important pathogen.

## Figures and Tables

**Figure 1 pathogens-13-00997-f001:**
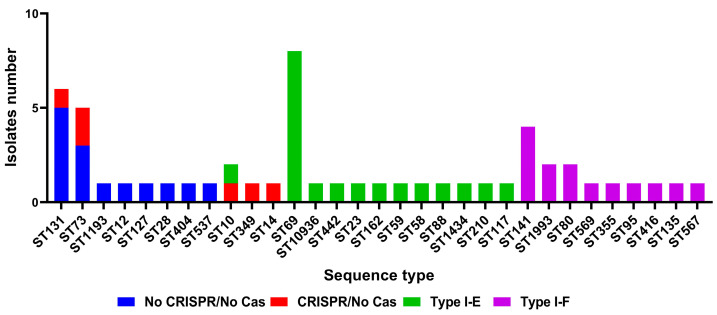
Sequence type distribution among different groups (“No CRISPR/No Cas”, “CRISPR/No Cas”, “Type I-E”, and “Type I-F”) of Crie-Pu *E. coli* isolates.

**Figure 2 pathogens-13-00997-f002:**
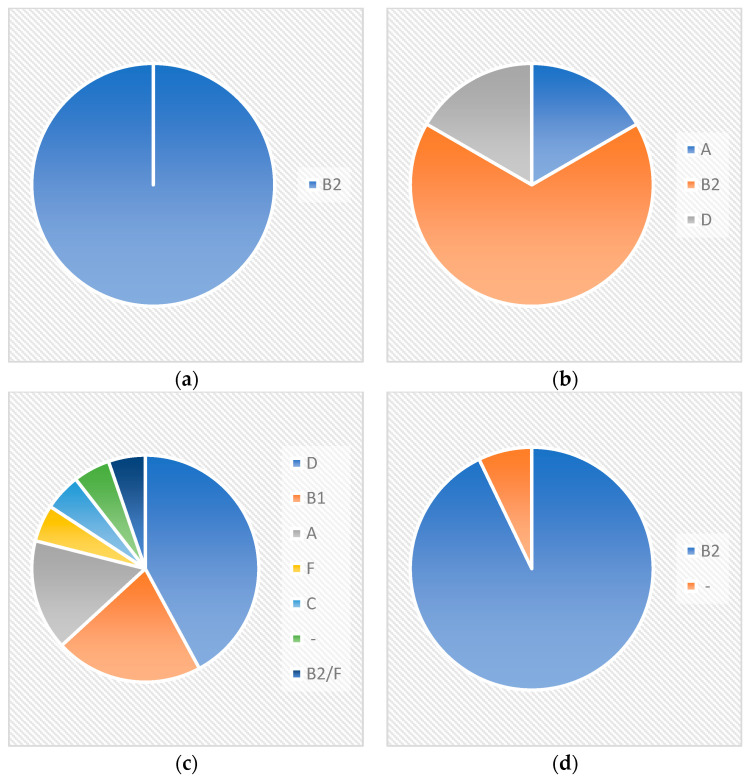
Phylogenetic groups’ distribution within different CRISPR-elements groups (“No CRISPR/No Cas”, “CRISPR/No Cas”, “Type I-E”, and “Type I-F”) for Crie-Pu *E. coli* isolates. (**a**) Crie-Pu isolates bearing neither CRISPR array, nor cas cassette; (**b**) Crie-Pu isolates with confirmed CRISPR arrays, but without *cas* cassettes; (**c**) Crie-Pu isolates with Type I-E CRISPR/Cas systems; (**d**) Crie-Pu isolates with Type I-F CRISPR/Cas systems.

**Figure 3 pathogens-13-00997-f003:**
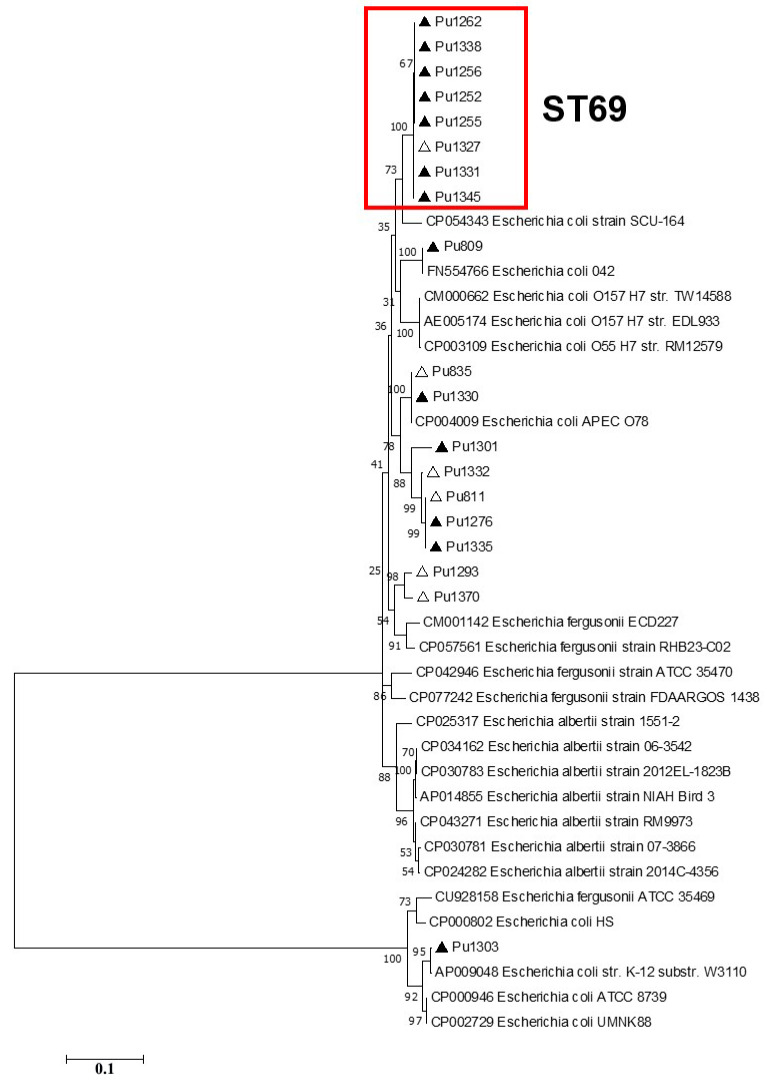
Maximum likelihood phylogenetic tree of the full-length Type I-E *cas3* gene sequences of clinical Crie-Pu *E. coli* isolates (shown as ‘Pu’ for brevity) and reference *Escherichia* isolates obtained from CRISPRCasdb. Bootstrap test (1000 replicates) was applied. Bootstrap values are indicated at the branch nodes. Antibiotic-resistant Crie-Pu *E. coli* isolates are marked with black triangles, antibiotic-sensitive Crie-Pu *E. coli* isolates are marked with white triangles. The genes identified in this study are indicated by the short isolate names, and the reference sequences are shown by GenBank accession number and strain name.

**Figure 4 pathogens-13-00997-f004:**
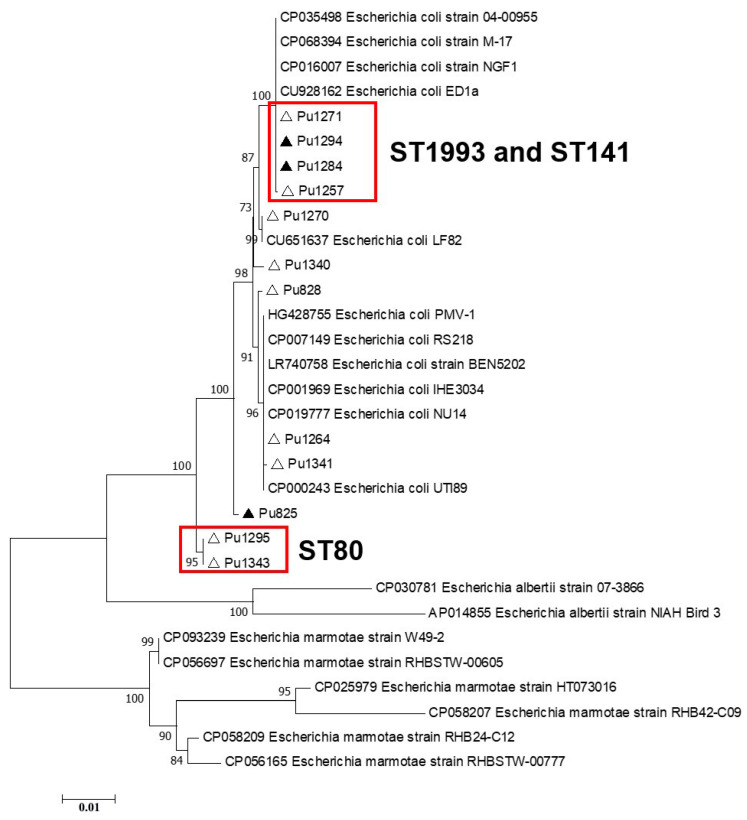
Maximum likelihood phylogenetic tree of the full-length Type I-F *cas3* gene sequences of clinical Crie-Pu *E. coli* isolates (shown as ‘Pu’ for brevity) and reference Escherichia isolates obtained from CRISPRCas database. Bootstrap test (1000 replicates) was applied. Bootstrap values are indicated at the branch nodes. Antibiotic-resistant Crie-Pu *E. coli* isolates are marked with black triangles, antibiotic-sensitive Crie-Pu *E. coli* isolates are marked with white triangles. The genes identified in this study are indicated by the short isolate names, and the reference sequences are shown by GenBank accession number and strain name.

**Figure 5 pathogens-13-00997-f005:**
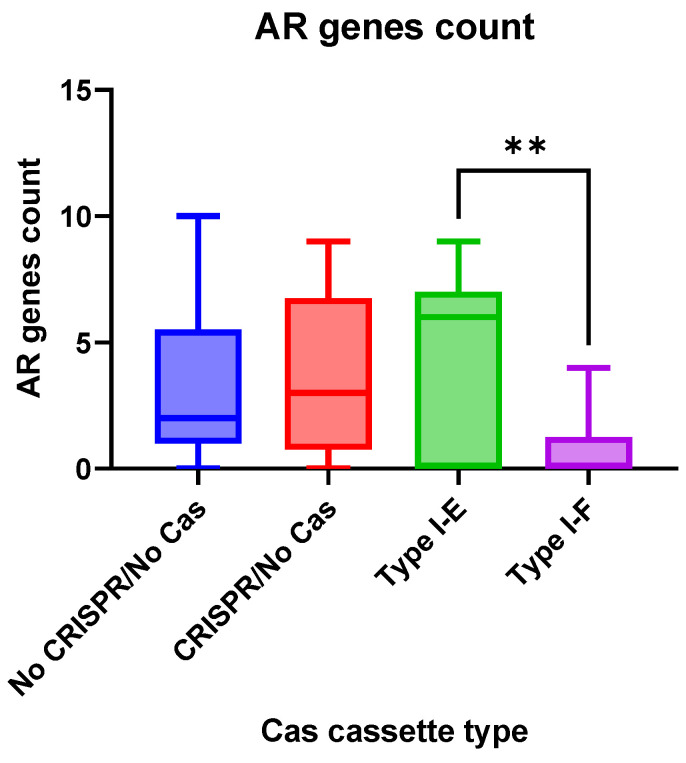
The number of antibiotic resistance genes in the analyzed groups of the Crie-Pu *E. coli* isolates. Asterisks denote significant difference (** *p* ≤ 0.01).

**Figure 6 pathogens-13-00997-f006:**
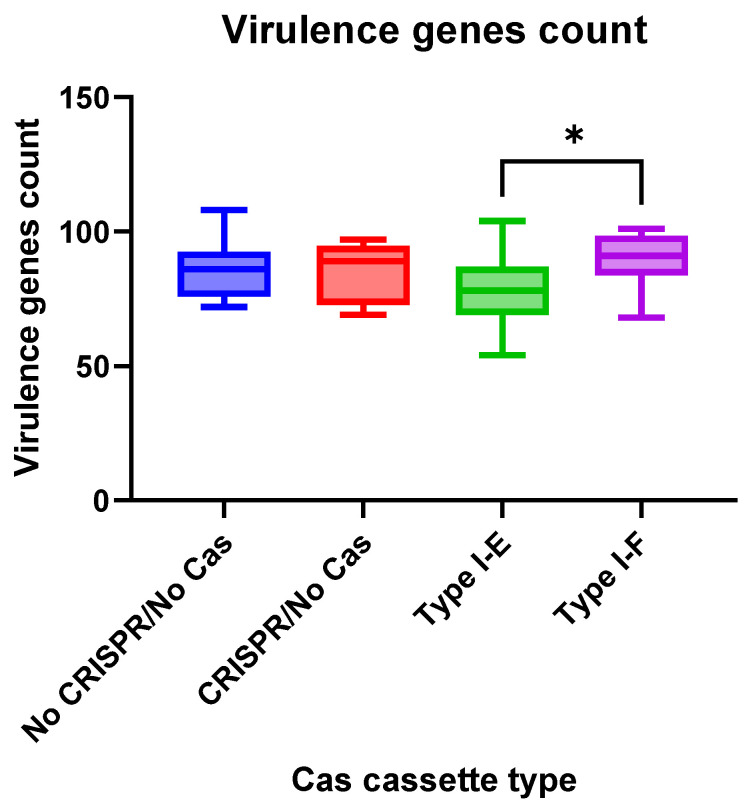
Virulence genes number in the analyzed groups of the Crie-Pu *E. coli* isolates. Asterisk denotes significant difference (* *p* ≤ 0.05).

**Figure 7 pathogens-13-00997-f007:**
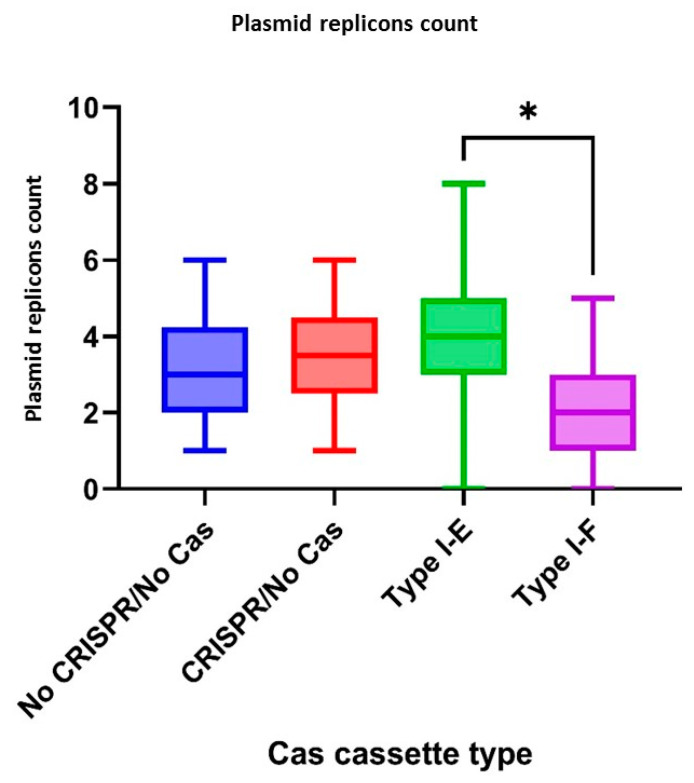
Plasmid replicons number in the analyzed groups of the Crie-Pu *E. coli* isolates. Asterisk denotes significant difference (* *p* ≤ 0.05).

**Figure 8 pathogens-13-00997-f008:**
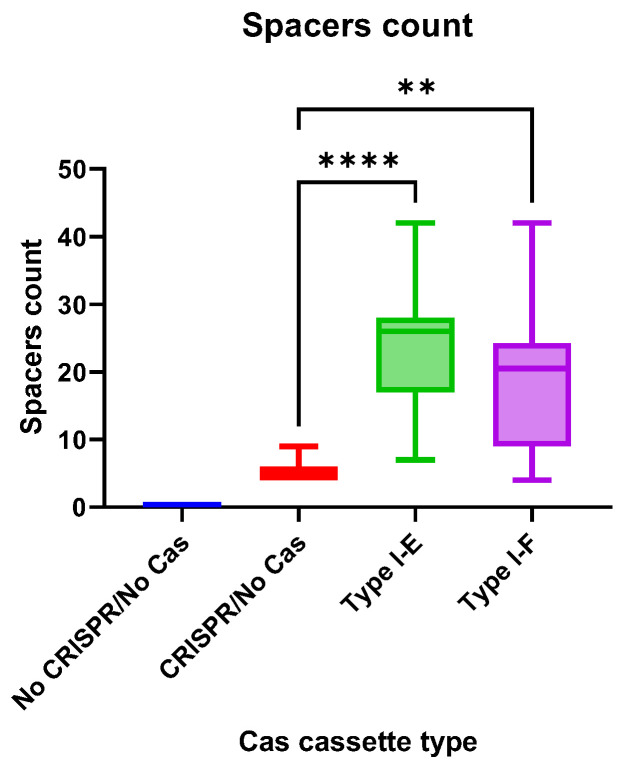
The CRISPR arrays spacers count in the analyzed groups of the Crie-Pu *E. coli* isolates. Asterisks denote significant differences (** *p* ≤ 0.01, **** *p* ≤ 0.0001).

**Figure 9 pathogens-13-00997-f009:**
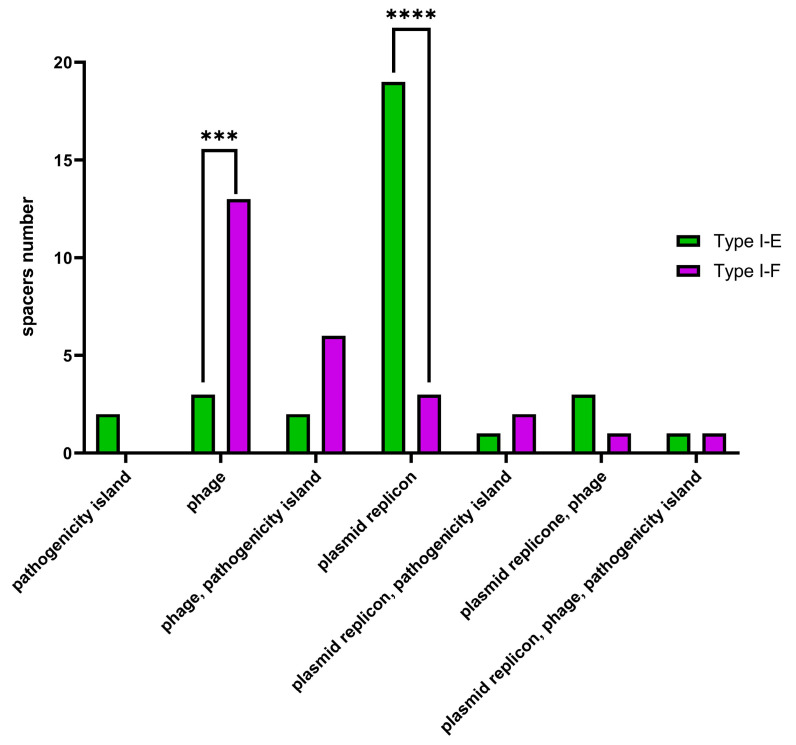
CRISPR array spacers distribution in Type I-E and Type I-F Crie-Pu *E. coli* isolates according to CRISPRTarget web service. Asterisks denote significant differences (*** *p* ≤ 0.001, **** *p* ≤ 0.0001).

**Table 1 pathogens-13-00997-t001:** Significant correlations in the analyzed groups of Crie-Pu *E. coli* isolates.

Correlation/Cas Type	No CRISPR/ No Cas	CRISPR/ No Cas	Type I-E	Type I-F
Antibiotic resistance gene count vs. Virulence factor gene count	---	---	---	---
Antibiotic resistance gene count vs. Plasmid count	---	---	r = 0.48, *p* = 0.039	r = 0.52, *p* = 0.060
Antibiotic resistance gene count vs. Spacer count	---	---	---	---
Virulence factors gene count vs. Plasmid count	r = −0.60, *p* = 0.025	---	---	---
Virulence factor gene count vs. Spacer count	---	---	---	---
Plasmid count vs. Spacer count	---	---	---	---

## Data Availability

The genomes of all isolates described in this study were uploaded to NCBI under the project number PRJNA1151703.

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
