# Peer review of "Interrelation Between Pathoadaptability Factors and Crispr-Element Patterns in the Genomes of Escherichia coli Isolates Collected from Healthy Puerperant Women in Ural Region, Russia"

_pathogens, 2024, doi:10.3390/pathogens13110997_

Round 1

Reviewer 1 Report

Comments and Suggestions for Authors

This study analyzes 53 E. coli isolates from healthy postpartum women, revealing diverse sequence types and phylogenetic groups, with most belonging to the B2 phylogroup. The data includes antibiotic resistance profiles, virulence factors, and plasmid replicons, contributing to further research on its pathogenicity and the role of CRISPR/Cas systems in its genome evolution. This manuscript meets the scope of Pathogens; therefore, it is recommended for acceptance after revision. The following are specific suggestions for modification.

1.     Lines 165, 168 and 180. The word “E. coil” should be italicized in the Results. Please check the entire manuscript, including references.

2.     Line 199, In addition to the detection quantity, please also provide the detection percentage. For example: type I-E, type I-F.

3.     Figure 1. Please redraw Figure 1, as some ST types cannot be found in the figure. It is necessary to label what the x and y axes represent.

4.     Using a heatmap to display both the clustering tree and the genetic information would enhance understanding. It is not appropriate to place so many results in the supplementary materials (Tables S2-S5).

5.     It is recommended to combine Figures 5-8 and label them as A, B, C, and D.

6.     Are there any other studies on isolates from pregnant women and mothers? If so, please disucuss.

7.     Please enrich the discussion on the CRISPR/No Cas types and phylogenetic trees.

8.     Please explain why the samples were concentrated in the B2.

Comments on the Quality of English Language

no comments

Author Response

We would like to thank the reviewer for the comments that led to significant improvement of our manuscript.

  1. Lines 165, 168 and 180. The word “E. coil” should be italicized in the Results. Please check the entire manuscript, including references.

Fixed as suggested

  1. Line 199, In addition to the detection quantity, please also provide the detection percentage. For example: type I-E, type I-F.

We added percentage values (lines 214-215 of the revised manuscript)

  1. Figure 1. Please redraw Figure 1, as some ST types cannot be found in the figure. It is necessary to label what the x and y axes represent.

Fixed as suggested

  1. Using a heatmap to display both the clustering tree and the genetic information would enhance understanding. It is not appropriate to place so many results in the supplementary materials (Tables S2-S5).

Tables S2, S4 and S5 were modified to heat map format; table S3 was simplified for better clarity and perception.

  1. It is recommended to combine Figures 5-8 and label them as A, B, C, and D.

Thank you for the comment, but we decided not to combine these figures. From our point of view, it is more clearly for readers to see the results as they are mentioned in the text.

  1. Are there any other studies on isolates from pregnant women and mothers? If so, please discuss.

      Such publications were mentioned in “Discussion” section; see References [29-31]

  1. Please enrich the discussion on the CRISPR/No Cas types and phylogenetic trees.

      We added comments in “Discussion” section on the mentioned data.

  1. Please explain why the samples were concentrated in the B2.

All of the isolates under investigation are of extra intestinal localization. Generally, E. coli isolates of extra intestinal origin are represented by B2 and D phylogroups.

Reviewer 2 Report

Comments and Suggestions for Authors

The aim of the present work was to provide a comprehensive analysis of 53 genomes of E. coli collected from healthy puerperal women in terms of CRISPRcas systems, plasmid replicons, virulence genes, antimicrobial susceptibility, antimicrobial resistance genes, phylogroups and ST-types distributions. Although on a limited number of genomes, authors found very interesting correlations which have the potential to pave the way to future understanding of the role of CRISPRcas system, not only in immunity but also in genome plasticity, evolution and stress adaptation. Only minor, yet relevant, concern arose:

1)    L97-98: no details are given on the protocol used to isolate and identify E. coli strains. Please add text.

2)    L283-286: authors reported that 7 fully susceptible E.coli isolates carried antimicrobial resistance associated genes. This seems an inconsistency between phenotype and genotype. These discrepancies are not infrequent but deserve a closer look. For example for one of those strains (CRIE-Pu 1328) the discrepancy is due to the fact that AMR detected genes are associated to antimicrobial agents not tested in the phenotypic susceptibility test: detected genes are related to resistance to b-lactams, trimethoprim, macrolides, fluoroquinolones and sulphonamides. However, no antimicrobial agent belonging to these antimicrobial classes was tested. The only exceptions are amoxicillin+clavulanic acid (b-lactams) and ciprofloxacin (fluoroquinolones). Crie-Pu 1328 was susceptible against amoxicillin+clavulanic acid. However being clavulanic acid an inhibitor of b-lactamases, without a test on amoxicillin alone, no susceptibility results against b-lactams can be ruled out. Moreover, Crie-Pu was susceptible to ciprofloxacin and carried the qnrB gene. This is not a discrepancy since it is well known that qnrB genes confer a phenotype of reduced susceptibility  rather than resistance to fluoroquinolones. Only the addition of point mutations in the QRDR region leads to full resistance. Therefore, I suggest to have a closer look to all 7 fully susceptible strains to see whether there is a true discrepancy or not between the AMR phenotype and genotype and then rephrase section L283-286.

3)    L497-504: One of the most interesting results is the negative correlation between Type I-F and AMR genes/plasmid replicons. Authors indicated Long et al., 2019  as reference with similar findings. In Long et al., 2019, the role of Type I-F in limiting the acquisition of AMR genes/plasmids was inferred from the observation of the homology between spacers and antimicrobial resistance plasmids. However in the present study, “plasmid” spacers were significantly lower in Type I-F systems in comparison to Type I-E. How authors explain this inconsistency? Please include this explanation also in the text.

Author Response

We would like to thank the reviewer for the comments that led to significant improvement of our manuscript.

  • L97-98: no details are given on the protocol used to isolate and identify E. coli strains. Please add text.

We added comments, section 2.1.

  • L283-286: authors reported that 7 fully susceptible E.coli isolates carried antimicrobial resistance associated genes. This seems an inconsistency between phenotype and genotype. These discrepancies are not infrequent but deserve a closer look. For example for one of those strains (CRIE-Pu 1328) the discrepancy is due to the fact that AMR detected genes are associated to antimicrobial agents not tested in the phenotypic susceptibility test: detected genes are related to resistance to b-lactams, trimethoprim, macrolides, fluoroquinolones and sulphonamides. However, no antimicrobial agent belonging to these antimicrobial classes was tested. The only exceptions are amoxicillin+clavulanic acid (b-lactams) and ciprofloxacin (fluoroquinolones). Crie-Pu 1328 was susceptible against amoxicillin+clavulanic acid. However being clavulanic acid an inhibitor of b-lactamases, without a test on amoxicillin alone, no susceptibility results against b-lactams can be ruled out. Moreover, Crie-Pu was susceptible to ciprofloxacin and carried the qnrB gene. This is not a discrepancy since it is well known that qnrB genes confer a phenotype of reduced susceptibility  rather than resistance to fluoroquinolones. Only the addition of point mutations in the QRDR region leads to full resistance. Therefore, I suggest to have a closer look to all 7 fully susceptible strains to see whether there is a true discrepancy or not between the AMR phenotype and genotype and then rephrase section L283-286.

We are very grateful to the reviewer for such a detailed comment.

Unfortunately, in terms of clarifying phenotype/genotype discrepancies we are limited to the panel of AMR compounds according to EUCAST. Obviously, it is very interesting and useful but it lies beyond the scope of the work presented. Presumably, it is going to be one of the focuses of our future investigations.

3)    L497-504: One of the most interesting results is the negative correlation between Type I-F and AMR genes/plasmid replicons. Authors indicated Long et al., 2019  as reference with similar findings. In Long et al., 2019, the role of Type I-F in limiting the acquisition of AMR genes/plasmids was inferred from the observation of the homology between spacers and antimicrobial resistance plasmids. However in the present study, “plasmid” spacers were significantly lower in Type I-F systems in comparison to Type I-E. How authors explain this inconsistency? Please include this explanation also in the text.

In the present study, we demonstrated that the number of AMR genes and plasmid replicons was higher in E. coli isolates bearing Type I-E CRISPR/Cas system than in isolates with Type I-F systems and this fact is consistent with observation published earlier [39]. In 2019, Long et al. have demonstrated the role of Type I-F system in limiting the acquisition of antimicrobial resistance [37]. Interestingly, the majority of our E. coli isolates carrying I-F system included fewer antimicrobial resistance genes compared to other isolates under investigation but were characterized by lower number of plasmid-targeting spacers than Type I-E CriePu E. coli isolates. This fact is in contrast to the work of Aydin S. et al., who reported a much larger proportion of plasmid-specific spacers in E. coli isolates susceptible to antimicrobial drugs [39]. This observation suggests more complex role of CRISPR/Cas systems in bacterial pathoadaptability and needs further investigations.

We included the above explanation in the “Discussion” section.